# The Current State of Veterinary Toxicology Education at AAVMC Member Veterinary Schools

**DOI:** 10.3390/vetsci9120652

**Published:** 2022-11-22

**Authors:** David C. Dorman, Robert H. Poppenga, Regina M. Schoenfeld-Tacher

**Affiliations:** 1Department of Molecular Biosciences, College of Veterinary Medicine, North Carolina State University, Raleigh, NC 27607, USA; 2California Animal Health and Food Safety Laboratory, School of Veterinary Medicine, University of California, Davis, CA 95616, USA

**Keywords:** curriculum, veterinary, toxicology, competency-based veterinary education, entrustable professional activities

## Abstract

**Simple Summary:**

Veterinary student education in toxicology is important. This study surveyed individuals affiliated with veterinary schools that are members of the Association of American Veterinary Medical Colleges. The online survey was designed to collect information about the credentials of the faculty that teach toxicology at veterinary schools, the topics that they include in their coursework, and faculty assessments regarding how prepared new graduates are at performing professional activities related to clinical toxicology. Nearly half of all schools provided a response to our survey. We found that toxicology was included as part of all veterinary school coursework. Toxicology classes across the different programs shared similar content. Most respondents to our survey felt that most of their students were prepared to perform professional activities related to veterinary toxicology.

**Abstract:**

This study assessed the depth, breadth, and perception of toxicology education in curricula at Association of American Veterinary Medical Colleges (AAVMC) member veterinary schools. An online questionnaire was sent twice to all 54 AAVMC members and sent once to a veterinary toxicology list serve. The survey covered areas related to instructor demographics, the depth and extent of toxicology taught, and the respondent’s perceptions of their student’s ability to perform entrustable professional activities (EPA). Results were analyzed using descriptive statistics. Our survey resulted in a 44% response rate. All responding schools included toxicology in their curriculum, and it was a required course in 23 programs. Contact hours in stand-alone veterinary toxicology courses ranged from 14 to 45 h. Most respondents indicated that the current time allotted for toxicology was inadequate, despite indicating that most of their students could perform most EPAs autonomously. One exception related to the ability of students to analyze toxicology data. We found small variations in teaching methods and curriculum content. The results of our study can assist veterinary schools in evaluating their curricula to better prepare new graduates for the management of toxicology issues they may face in their veterinary careers.

## 1. Introduction

Toxicology is an experimental science that investigates the fate of chemicals in the body, the mode of action of chemicals, and the adverse effects they induce. Veterinary toxicology has both basic (e.g., comparative toxicology) and clinical (e.g., veterinary clinical toxicology) science components. In many veterinary school curricula, students initially focus on the basic sciences, while later in their training, the focus shifts to clinical sciences. Thus, toxicology instruction, like training in pharmacology [1], should span these stages of veterinary education.

Domestic animals and wildlife can be exposed to a vast array of potential toxicants, including human and veterinary medications; illicit drugs; toxic plants, mycotoxins, algal toxins, and microbial toxins; pesticides; toxic metals and metalloids; and a variety of household products and industrial chemicals. The effects of toxicants can vary amongst different animal species. For example, exposure to the mycotoxin fumonisin is associated with equine leukoencephalomalacia and a pulmonary edema syndrome in swine [2]. The diversity of animal species that may be exposed to these agents and important species differences in response distinguishes veterinary from human toxicology.

Core concepts in toxicology as well as the diverse set of potentially toxic agents and animal species of interest to veterinarians make the design of a curriculum in veterinary toxicology challenging. Information regarding the current state of the toxicology curricula in professional veterinary programs in the United States and elsewhere is scarce. Toxicologists engaged in education and affiliated with either the American Board of Veterinary Toxicology (ABVT) or the American Academy of Veterinary and Comparative Toxicology developed a priority listing of fundamental concepts and knowledge that entry-level veterinarians should possess [3]. This included nine concepts (e.g., biotransformation, diagnosis, dose–response) and 73 agents (e.g., acetaminophen, botulism, lead) or classes of agents (e.g., cardiac glycoside-containing plants, ionophores, mycotoxins) with various effects in different animal species. The methods used in identifying and prioritizing these agents were not provided. This proposed curriculum did not consider the amount of time needed for delivery. A conservative estimate that each core concept could require one hour of instruction and each agent or agent class another 15 min yields a total of approximately 27 h of instruction. Development of toxicology curricula at veterinary schools could also be informed by efforts in human toxicology. Consensus-based lists of core topics that should be included in curricula for medical toxicology clerkships in medical schools have been proposed [4,5,6]. One proposed curriculum includes over 200 agents or classes of agents in addition to an extensive consideration of core concepts [5].

Veterinary education is increasingly focusing on assessing essential competencies in which trainees must become proficient before undertaking them independently [7]. To that end, a team of educators has proposed eight entrustable professional activities (EPAs) for the assessment of workplace activities deemed essential for veterinary practice [8,9]. These have been proposed as a practical framework for the implementation of competency-based education. An EPA is defined as “an essential task of a discipline that a learner can be trusted to perform with limited supervision in a given context and regulatory requirements, once sufficient competence has been demonstrated” [9]. The core EPAs help define Day One practice expectations for every veterinary graduate.

The ABVT recently formed an Education Committee to better inform the specialty on ways that veterinary toxicology is currently being taught at Association of American Veterinary Medical Colleges (AAVMC) member veterinary schools. Colleges and schools of veterinary medicine that seek to join the AAVMC must be accredited by the American Veterinary Medical Association (AVMA) Council on Education (COE). There are currently 54 AAVMC members, including all 33 veterinary colleges in the United States. This study helps address current gaps in our knowledge and provides an updated report on veterinary toxicology education in the United States and elsewhere.

## 2. Materials and Methods

An online survey consisting of 34 items was created to assess the extent and depth of toxicology education in veterinary programs at AAVMC member schools. The survey was reviewed and approved by both the North Carolina State University Institutional Review Board (IRB) and the AAVMC Survey Committee. The survey was developed by the authors and tested by several individuals prior to distribution to determine completion time and identify components that needed revision or clarification. The complete survey is provided in Appendix A. Participation in the study was voluntary, and all participants received descriptive information regarding the study along with a link to the survey. All invitees were informed that their information would be kept confidential.

The survey was distributed twice (on 6 July 2021 and 16 November 2021) by the AAVMC Survey Committee via email to the Associate Deans of Academic Affairs at member institutions. Distribution occurred using a web-based Qualtrics survey tool. The solicitation email to the Associate Deans requested them to forward the survey invitation to the faculty member most involved in teaching toxicology in their veterinary curriculum. The survey was also distributed (on 28 July 2022) to individuals that participate in the ABVT email listserv. The survey was kept open for approximately 13 months and was closed on 16 September 2022.

The first survey question addressed informed consent. An affirmative response was required to proceed with the rest of the questionnaire. Survey questions were divided into distinct sections, including portions focused on consent, instructor data, veterinary curriculum, veterinary courses, opinions, curricular change, and syllabi. Instructor data collected demographic information about the program and the credentials of individuals providing instruction in veterinary toxicology. The section on the toxicology curriculum asked respondents to estimate the amount of toxicology instruction provided by different types of individuals (e.g., faculty members, trainees, adjunct faculty). Data sought about veterinary courses included course titles, methods of instruction, whether courses were required or electives, resources used, and course content. This section also asked the respondent’s perception of how prepared their veterinary students would be at the time of graduation to perform the seven entrustable professional activities (EPAs) pertaining to a case involving toxicant exposure. It also asked respondents whether they would use peer-reviewed online materials to either teach or assess these EPAs. Respondent opinions regarding the adequacy of the toxicology curriculum were collected. The next section addressed veterinary curriculum changes at the respondent’s institution and student interests in toxicology as a career. The final section provided respondents with the opportunity to upload syllabi for their required or elective toxicology courses. All results were analyzed using descriptive statistics. Mean (±standard deviations) have been provided for some response data.

## 3. Results

A total of 29 surveys were received. Three invalid surveys (incomplete demographic data including identification of the institution) were excluded from our study. One completed survey came from an institution that was not a member of the AAVMC, and data from this survey were excluded from most analyses. This survey response did provide a toxicology course syllabus that contributed to the list of agents presented in Appendix B. Of the 54 schools and colleges eligible for the study, 24 unique institutions responded to the survey, resulting in an overall 44% response rate. Most (*n* = 18/24) responses were from academic institutions based in the United States. Three responses were obtained from institutions in Australia, England, or New Zealand, two responses were received by Canadian schools, and one response was obtained from a school based in the Caribbean. Two responses were received from faculty at one veterinary school based in the United States. These individuals were responsible for different facets of the toxicology curriculum, and their individual responses were retained as appropriate, resulting in 25 respondents to the survey. Most (*n* = 16/18) of the veterinary schools based in the United States and Canada were affiliated with academic institutions that also had a diagnostic laboratory accredited by the American Association of Veterinary Laboratory Diagnosticians (AAVLD). Most (*n* = 23/25) of the respondents personally teach toxicology to veterinary students at their home institution. The mean (±SD) number of years that the respondents had taught toxicology at their home institution and the total number of years they had taught toxicology across all locations were 12.7 ± 9.9 and 20.1 ± 10.3 years, respectively.

Of the 23 participants reporting their credentials, most (*n* = 20) respondents held a veterinary degree (DVM or equivalent). Other degrees held by respondents included a Master’s degree (*n* = 1) or Ph.D. or equivalent (*n* = 15). Board certification was common among the respondents. Eleven individuals were certified by the ABVT, four by the American Board of Toxicology, two by the American College of Veterinary Pathologists, one by the American College of Theriogenology, one by the American College of Veterinary Internal Medicine, one by the American College of Veterinary Emergency Critical Care, and one European Registered Toxicologist. Most (*n* = 18/24) of the AAVMC member schools relied on a single individual to teach the majority of the toxicology in the curriculum. However, most of the veterinary programs had additional individuals teach more than two hours of contact time in the veterinary toxicology curriculum. The mean number of instructors involved in teaching toxicology at AAVMC member institutions was 3.6 ± 2.8 (range 1–10). Most of the toxicology curriculum was provided by faculty members at the home institution. Trainees or staff provided some instruction at four schools (1–45% of the contact time in toxicology). Two veterinary schools relied almost entirely (>90% instruction time) on either adjunct/locum instructors or guest lecturers. A third institution indicated that 35% of the instruction in toxicology was provided by an adjunct/locum instructor.

Table 1 provides a summary of the method of instruction used prior to the COVID-19 pandemic and the types of educational resources that are currently used to teach toxicology to veterinary students. Only 25% (*n* = 6/24) of the respondents indicated that recent changes in teaching format due to the COVID-19 pandemic would cause a permanent alteration in the way toxicology was taught in the future. Most respondents (*n* = 16/24) indicated that a permanent change was possible because of recent changes in teaching format due to the COVID-19 pandemic. Most respondents (*n* = 19/24) indicated that a curriculum change had occurred within the past ten years at their home institution. No change in contact time because of a curriculum change was reported by most of the respondents (*n* = 16/23). Only one respondent indicated that contact time in toxicology increased following a curriculum change at their institution. Fifteen respondents provided comments regarding how they would use additional time to teach toxicology. The most frequently mentioned ways additional time dedicated to toxicology would be used by these 15 respondents included increased time allocated to cases (66.7%), enhanced discussion of the management of poisoned animals (33.3%), increased coverage of poisonous plants (33.3%), and discussion of topics related to environmental and ecotoxicology (20%).

Information about toxicology courses was provided for 22 institutions. All survey respondents indicated toxicology instruction is provided within their veterinary curriculum. Toxicology as a stand-alone course or as part of a broader course (e.g., veterinary pharmacology and toxicology, pathobiology) was required at nearly all programs (*n* = 21/22). Additional elective or selective courses in toxicology were available at some (*n* = 7/22) of the responding institutions. Mean contact hours in stand-alone veterinary toxicology courses was 30.5 ± 10.7 h (range: 14–45 h, *n* = 12). Most respondents (*n* = 17/24) felt that more time was needed to prepare students for day-one competency.

Our survey found that toxicology as a career option is considered by some veterinary students. The percentage of students in each class indicating an interest in toxicology, however, varies between institutions. Most (17/23) respondents indicated that fewer than 5% of their veterinary students expressed an interest in toxicology as a career choice. Two respondents however indicated that over 20% of their students expressed an interest in toxicology as a career.

Toxicology topics taught at AAVMC member institutions are reported in Table 2. Most topics were taught using a comparative approach that considered multiple species. Because veterinary toxicology is often taught in multiple courses, many toxicology topics were also included in courses that focused on either small animals, horses, large animals, or wildlife and exotics. One topic that appears less frequently is environmental toxicology. Likewise, toxicology in wildlife and exotic species is also less commonly included in existing veterinary curricula. This observation was also borne out by comments made by several respondents.

Respondent assessment of the competency of their students to perform entrustable professional activities (EPA) is provided in Table 3. In most cases, respondents felt that > 80% of their students could perform most toxicology-related EPAs autonomously and in some cases could mentor others in an activity. One exception related to the ability of students to analyze toxicology data where only 50% of respondents felt their students could perform this activity without supervision. A related question queried the confidence of the respondents in their ability to evaluate students’ preparation to perform an EPA. Approximately 10 to 20% of respondents indicated that they lacked confidence in their ability to evaluate their student’s ability to perform one or more EPAs.

Respondents were generally receptive to using online, peer-reviewed materials to either teach or assess EPAs (Table 4).

A total of 22 respondents provided comments related to perceived strengths and weaknesses in their institution’s toxicology curriculum. Multiple respondents reported similar strengths in their curriculum, including broad coverage of agents and general principles of toxicology in the curriculum (54.5%), use of a case- or problem-based approach to presenting the curriculum (27.3%), toxicology was taught by one or more experienced veterinary toxicologists (18.5%), and availability of a toxic plant garden (13.6%). Several respondents viewed a stand-alone toxicology course as a strength of the program. These respondents also provided comments related to programmatic gaps or weaknesses. Overall, these responses varied between institutions with fewer common concerns. Several of the 22 respondents (18.2%) were concerned about the lack of integration of toxicology in the overall curriculum, including an overall lack of knowledge of how toxicology may be taught by others in the curriculum. Likewise, 18.2% of respondents noted a lack of time dedicated to the teaching of toxicology. Two instructors with specialization in critical care or pathology indicated that they may not have the background needed to teach toxicology. There was also a concern by several respondents that topics related to forensics or large animal toxicology may not be adequately covered in the toxicology curriculum.

Syllabi for one or more courses were provided by 11 institutions. These included syllabi for required as well as elective courses. All required toxicology courses included one or more lectures on basic concepts (e.g., terminology, decontamination procedures, pharmacokinetics). Most syllabi also provided learning objectives. A summary of basic concepts and learning objectives found in multiple syllabi is provided in Appendix C. The remainder of the required courses often focused on toxicologic syndromes associated with toxic agents or classes of agents. Presentation of this material varied amongst institutions. Several syllabi used a systems approach (e.g., neurotoxicants) to organize the course, while others used an agent-based approach.

## 4. Discussion

Despite concerns about the demise of toxicology as a theme in the professional veterinary medical curriculum, all survey respondents indicated toxicology instruction is provided within their veterinary curriculum. Indeed, many AAVMC member institutions provide students with both required and elective courses in toxicology. The teaching of veterinary toxicology is generally provided by veterinarians, many of whom also hold advanced graduate degrees and additional board specialization. Nearly half of all institutions provide a stand-alone course in toxicology. This finding differs from that seen in United States and Canadian medical schools, where less than 5% had formal courses in toxicology [10]. Dedicated veterinary toxicology courses at AAVMC member institutions had between 14 and 45 contact hours. Mean contact hours devoted to toxicology training in American and Canadian medical schools were 5 and 6 h, respectively [10].

Based on our results, veterinary toxicology is often offered through a team-based education effort provided by multiple instructors. In fewer cases, the veterinary curriculum rests on the efforts of a single instructor. Prior to the COVID-19 pandemic, most instruction occurred in person with less reliance on online instruction. We examined this timeframe since the COVID-19 pandemic prompted instruction at many veterinary schools to switch to remote teaching formats. Most respondents thought that recent changes in teaching format due to COVID-19 could possibly change the way toxicology was taught. A recent study found student performance in a veterinary toxicology course delivered online asynchronously during the pandemic was similar to achievement during the prior, in-person course [11]. All respondents presented didactic lectures and provided students with copies of their PowerPoint presentations, suggesting this remains the primary means of instruction in veterinary toxicology. Case-based materials and problem sets were also used by many instructors. Our perception is that novel educational strategies, including flipped classroom techniques, are less commonly used in veterinary student instruction in toxicology. Approaches like flipping the classroom have been associated with improved academic performance in medical students [12].

Our survey explored the respondent’s assessment of their students’ ability to perform EPAs at graduation. The seven EPAs included in our study were modified from those developed by the AAVMC Competency-Based Veterinary Education Working Group [13]. These EPAs describe the most relevant activities a veterinarian carries out in private veterinary practice [14]. The EPAs provide an insight into what newly graduated veterinarians should be able to do and how much supervision may be required as they start professional practice [15]. Our data indicate that most (>80%) respondents felt their graduates could autonomously perform five of the seven EPAs included in our study. Approximately 25% of the respondents indicated that their graduates would require support to formulate recommendations for preventive healthcare. In retrospect, interpretation of this EPA could vary between respondents. For example, some individuals may perceive this to mean client communication regarding toxicologic hazards. Other respondents may have assumed this related to the initial management of an exposed animal. Mentorship was also needed for students to analyze toxicology data. This later finding is not unexpected since this represents an activity often associated with working toxicologists [16]. Interestingly, nearly one-quarter of all respondents lacked the confidence to assess their student’s ability to perform this EPA. Most (>80%) respondents were somewhat confident or very confident in their ability to assess the ability of their students to perform the remaining EPAs. Nearly 45% of all respondents indicated their graduates would require guidance to interpret toxicology data. The majority (>50%) of respondents indicated that they would be likely to use online, peer-reviewed materials to either teach EPAs or assess their students’ ability to perform an EPA. Less than 10% of all respondents indicated that they were unlikely to use these resources if they were available.

Based on our results, the majority of respondents used a comparative approach to teaching veterinary toxicology to their students. With few exceptions, there were common topics incorporated into veterinary curricula across responding institutions. One such exception related to wildlife and ecotoxicology was mentioned several times as a weakness of the toxicology curriculum. Our finding that major concepts and topics are often shared across different academic disciplines suggests that a next step could involve the definition of an agreed-upon set of core concepts for veterinary toxicology curricula. Similar approaches have occurred for pharmacology [17,18,19] and toxicology [20] programs within human health professions education, as well as for farm animal medicine [21] within veterinary medicine. Evidence from biology and pharmacology education indicates that core concepts are useful and effective structures around which such a curriculum can be designed to facilitate student learning [22,23]. Subsequent efforts could focus on the selection and development of specific assessment instruments to measure students’ ability to perform the EPAs to demonstrate ability to manage cases of toxicant ingestion, building on the efforts of Duijin and coworkers [24]. A group of veterinary educators under the auspices of the CBVE Working Group have proposed a draft Competency-Based Veterinary Education Toolkit, outlining recommended assessment formats for each CBVE Domain of Competency [25]. Among these, Script Concordance Tests (SCTs) stand out as a method faculty could use to assess their students’ ability to analyze toxicology data [26,27]. Since these examinations are challenging and time consuming to develop and validate, the fact that most curricula include similar toxicology content and teaching methods paves the way for a consortium effort to develop assessment tools.

There are methodological limitations to our study. Our data are often derived from a single individual at each school and may be biased because either the respondents have vested interests in the survey outcome, or they interpret the questions differently. Another major limitation of this investigation is the response rate. Acceptable response rates for web-based surveys remain poorly defined. Our response rate (44%) is qualitatively similar to that seen in a meta-analysis that reported a mean response rate for web surveys was 39.6% [28]. It remains uncertain if those who responded to our survey have similar views regarding veterinary toxicology as those who did not respond. Despite our effort to increase the response rate by careful consideration of the number of survey questions, using an easily accessible web-based survey, and sending an email reminder, the response rate was still only moderate. We believe that many factors contributed to our low rate, including survey fatigue, time constraints, and the time required to finish the survey. These factors have been noted by others [29]. Distribution of the survey relied on Associate Deans of Academic Affairs forwarding the email they received from the AAVMC. Associate Deans and recipient faculty may have missed these email communications resulting in decreased response rates. An additional plausible explanation is that some survey recipients may not have had the appropriate knowledge about toxicology instruction status at their institution. Despite the moderate response rate, our study provides valuable information on the current perception of the respondents regarding toxicology instruction in veterinary programs.

## 5. Conclusions

In conclusion, this study describes the current depth and breadth of toxicology instruction in a sample of veterinary programs at AAVMC member schools. There was a general agreement among the respondents that veterinary schools and colleges devote curricular space to toxicology instruction. The depth and the extent of toxicology topics coverage varied among institutions.

## Figures and Tables

**Table 1 vetsci-09-00652-t001:** Current didactic resources and primary method of instruction provided in toxicology courses between 2018 and 2019 at AAVMC member institutions (*n* = 24). Percentage of institutions.

Method of Instruction	%	Didactic Resources	%
In-person lecture	100.0	Printouts of instructor-created PowerPoints	100
In-person discussion/small group activities	62.5	Optional textbook(s)	79.2
In-person laboratory	37.5	Case-based materials	75.0
Asynchronous online instruction	12.5	Instructor-generated text-based notes	58.3
Field investigations	12.5	Practice quizzes	54.2
Synchronous online instruction	8.3	Problem sets	37.5
		Required textbook(s)	4.2

**Table 2 vetsci-09-00652-t002:** Basic veterinary toxicology concepts and topics taught at AAVMC member institutions (*n* = 24).

	% of Respondents				
Domain	All Species Comparative Perspective	Small Animal	Equine	Food Animal	Wildlife and/or Exotics
Clinical and diagnostic toxicology	87.0	17.4	17.4	17.4	8.7
Drugs	87.5	20.8	12.5	12.5	8.3
Environmental toxicology	65.0	10.0	10.0	15.0	35.0
Feed and water contaminants	72.7	22.7	18.2	27.3	18.2
Household chemicals	69.6	39.1	4.4	4.4	0
Metals and micronutrients	82.6	26.1	13.0	26.1	13.0
Mycotoxins	79.2	25.0	20.8	25.0	4.2
Pesticides	78.3	30.4	21.7	30.4	17.4
Poisonous plants	81.8	27.3	22.7	27.3	4.6
Principles of toxicology	95.7	8.7	8.7	8.7	0
Therapeutic measures	90.9	18.2	18.2	18.2	4.6
Toxic gases	76.5	5.9	5.9	23.5	5.9
Zootoxins	85.0	15.0	15.0	20.0	0

**Table 3 vetsci-09-00652-t003:** Respondent perceptions regarding entrustable professional activities associated with toxicology (number of responses varies per question).

	Respondent’s Assessment of Their Confidence in Their Students’ Ability to Perform an EPA at Graduation (%)				Respondent’s Confidence in Their Ability to Evaluate Students’ Preparation to Perform an EPA (%)		
Entrustable Professional Activity (EPA)	Not Competent	Can Perform with Support from a Mentor	Can Perform Autonomously	Can Mentor Others in Developing the Skill	Not Confident	Somewhat Confident	Very Confident
Recognize a patient requiring urgent or emergent care and initiate evaluation and management (*n* = 23)	0.0	13.0	82.6	4.4	12.5	33.3	54.2
Interpret toxicology data (*n* = 22)	9.1	40.9	50.0	0	16.7	50.0	33.3
Gather a toxicologic history, perform an examination, and create a prioritized differential diagnosis list (*n* = 23)	4.4	8.7	78.3	8.7	16.7	37.5	45.8
Formulate relevant questions and retrieve evidence to advance care (*n* = 23)	0	13.0	82.6	4.4	8.3	50.0	41.7
Formulate recommendations for preventive healthcare (*n* = 21)	4.8	14.3	66.7	14.3	20.8	45.8	33.3
Develop and implement a management/treatment plan (*n* = 23)	4.4	13.0	78.3	4.4	16.7	37.5	45.8
Develop a diagnostic plan and interpret results (*n* = 22)	4.6	13.6	72.7	9.1	12.5	33.3	54.2

**Table 4 vetsci-09-00652-t004:** Respondent’s likely use of online, peer-reviewed materials to either teach or assess EPAs (number of responses varies per question).

	Teach EPAs (%)			Assess EPAs (%)		
Entrustable Professional Activity (EPA)	Likely	May or May not	Unlikely	Likely	May or May not	Unlikely
Recognize a patient requiring urgent or emergent care and initiate evaluation and management (*n* = 23)	54.2	41.7	4.2	54.2	41.7	4.2
Interpret toxicology data (*n* = 22)	54.2	41.7	4.2	54.2	37.5	8.3
Gather a toxicologic history, perform an examination, and create a prioritized differential diagnosis list (*n* = 23)	62.5	25.0	12.5	54.2	37.5	8.3
Formulate relevant questions and retrieve evidence to advance care (*n* = 23)	58.3	33.3	8.3	50.0	45.8	4.2
Formulate recommendations for preventive healthcare (*n* = 21)	50.0	45.8	4.2	54.2	41.7	4.2
Develop and implement a management/treatment plan (*n* = 23)	66.7	20.8	12.5	58.4	33.3	8.3
Develop a diagnostic plan and interpret results (*n* = 22)	62.5	25.0	12.5	54.2	37.5	8.3

## Data Availability

Data is contained within the article or Appendix.

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
