# Peer review of "The Current State of Veterinary Toxicology Education at AAVMC Member Veterinary Schools"

_vetsci, 2022, doi:10.3390/vetsci9120652_

Round 1
Reviewer 1 Report
This study aimed to evaluate the toxicology subject in veterinary medicine education. Data collection is based on an online survey that included veterinary colleges that are members of the Association of American Veterinary Medical Colleges. The design of the study is logical and the article is systematically written.
Author Response
Thank you very much for your comments. No changes requested.

Reviewer 2 Report
This is a detailed description of the toxicology instruction in the veterinary medicine curriculum. Although the average response rate still provides insight into the status and lays a foundation for future improvement.
Author Response
Thank you very much for your feedback. No actions listed.
Reviewer 3 Report
This paper assesses the state of toxicology education in AA-VMC member veterinary schools. The survey was well-constructed and the analysis of the outcome was well-performed. Overall, this paper is an important contribution that provides a foundation for future work in veterinary toxicology education.
Also of interest is Appendix B which lists the toxic agents covered in one or more veterinary courses.
I was particularly excited to learn about the proposed "core concepts" model building on other work. In future work, it would be interesting to see how proposed core concepts could align with the Entrustable professional activities.
Comments:
In the abstract, there is a disconnect between simple summary stating that the respondants felt their students were able to perform professional activities related to vet tox, but then the abstract states that 14-45 hours were not adequate for teaching tox. This might be two different survey questions resulting in dissonance, but it isn't clear from the writing.
line 207 - sentence fragment.
Author Response
Abstract edited in lines 28-29 to make the discrepant results more clear.
Line 207 edited to correct sentence fragment.
